# Improved characterisation of MRSA transmission using within-host bacterial sequence diversity

Matthew D Hall[1]*, Matthew TG Holden[2], Pramot Srisomang[3], Weera Mahavanakul[4], Vanaporn Wuthiekanun[5], Direk Limmathurotsakul[5], Kay Fountain[6], Julian Parkhill[7], Emma K Nickerson[8], Sharon J Peacock[9], Christophe Fraser[1]

[1]Big Data Institute, University of Oxford, Oxford, United Kingdom; [2]School of Medicine, University of St Andrews, St Andrews, United Kingdom; [3]Department of Pediatrics, Sunpasitthiprasong Hospital, Ubon Ratchathani, Thailand; [4]Department of Medicine, Sunpasitthiprasong Hospital, Ubon Ratchathani, Thailand; [5]Mahidol Oxford Tropical Medicine Research Unit, Mahidol University, Salaya, Thailand; [6]Department of Biology and Biochemistry, University of Bath, Bath, United Kingdom; [7]Department of Veterinary Medicine, University of Cambridge, Cambridge, United Kingdom; [8]Department of Infectious Diseases, Cambridge University Hospitals NHS Foundation, Cambridge, United Kingdom; [9]Department of Medicine, University of Cambridge, Cambridge, United Kingdom

**Abstract** Methicillin-resistant *Staphylococcus aureus* (MRSA) transmission in the hospital setting has been a frequent subject of investigation using bacterial genomes, but previous approaches have not yet fully utilised the extra deductive power provided when multiple pathogen samples are acquired from each host. Here, we used a large dataset of MRSA sequences from multiply-sampled patients to reconstruct colonisation of individuals in a high-transmission setting in a hospital in Thailand. We reconstructed transmission trees for MRSA. We also investigated transmission between anatomical sites on the same individual, finding that this either occurs repeatedly or involves a wide transmission bottleneck. We examined the between-subject bottleneck, finding considerable variation in the amount of diversity transmitted. Finally, we compared our approach to the simpler method of identifying transmission pairs using single nucleotide polymorphism (SNP) counts. This suggested that the optimum threshold for identifying a pair is 39 SNPs, if sensitivities and specificities are equally weighted.

*For correspondence:
matthew.hall@bdi.ox.ac.uk

Competing interests: The authors declare that no competing interests exist.

## Introduction

Whole-genome sequencing (WGS) has proved a valuable tool in the investigation of the transmission of infectious agents. *Staphylococcus aureus*, and in particular methicillin-resistant *S. aureus* (MRSA) has been a frequent subject, often in a hospital environment where it is responsible for a considerable burden of disease. The focus of previous studies has ranged from identifying bacterial spread between continents (*Harris et al., 2010*), to reconstructing the detailed timelines of outbreaks in single hospital wards (*Köser et al., 2012*; *Harris et al., 2013*). Several prospective studies have also been conducted in single settings (*Price et al., 2014*; *Nübel et al., 2013*).

In a previous study, *Tong et al. (2015)* acquired and sequenced MRSA isolates from the high-transmission setting of two intensive care units (ICUs) in a Thai hospital. All were of sequence type 239 (ST 239). They showed that bacterial genetic diversity in this single location provided evidence

of at least five circulating clades in a state of temporal flux, with multiple clades colonising different subjects over the course of the study period. By sequencing a very large number of isolates from a single subject, they also demonstrated the existence of a considerable within-host 'cloud of diversity'. As a result, they cautioned against using a single pathogen genome as a representative of the colonisation of a single subject.

In the time since the publication of that study, methodological work has emerged to demonstrate how phylogenetic reconstruction using multiple genomes per subject can be used to identify individuals involved in pathogen transmission or colonisation events, and to reconstruct the direction of that transmission (*Romero-Severson et al., 2016*). The software package *phyloscanner* (*Wymant et al., 2017*) was subsequently developed to perform analyses of this kind. Thus there are now additional reasons to perform multiple sampling beyond the cautionary point made by Tong et al.

An obvious extension of multiple sampling is to sample from different body sites of the same host, and from there investigate the dynamics of colonisation at the within-host level. The anterior nares are the primary ecological niche for human *S. aureus* colonisation (*Kluytmans et al., 1997*), but the sensitivity of detection of MRSA carriage based on nasal screening alone is only 68–75% (*McKinnell et al., 2013*). Colonisation of multiple anatomical sites excluding the nose, however, appears rare (*Baker et al., 2010*; *Eveillard et al., 2006*) and eradication of nasal *S. aureus* is often followed by disappearance of the bacterium from other sites (*Parras et al., 1995*; *Reagan et al., 1991*). The process by which *S. aureus* spreads between sites has been little studied.

Another potential investigation is the size of the transmission bottleneck of an MRSA strain during the colonisation of a given individual, in other words the quantity of genetic diversity that is passed from one subject to another at transmission. This is an important concern when transmission routes are to be reconstructed using pathogen genomes, but inference is challenging in the absence of previous studies of this nature for *S. aureus*. Most previous work has not had access to dense within-host sampling and has used one sequence per host, which severely limits what can be determined about the bottleneck. Recently, *Worby et al. (2017)* demonstrated that shared genomic variants, identified using deep sequencing data, can be a powerful tool in identifying transmission pairs, but conclude that they are much more useful where the bottleneck is wide, as shared variants are rare if it is narrow. Selection of a suitable method for identifying pairs therefore requires quantification of the bottleneck width.

As the availability of multiple samples per host in standard microbiological practice is limited, genetic studies of MRSA outbreaks have, at least until recently, usually used only a single pathogen sequence per patient (*Harris et al., 2010*; *Harris et al., 2013*; *Nübel et al., 2013*; *Köser et al., 2012*). The investigation of transmission between individuals has then usually relied on selecting a threshold for the number of single nucleotide polymorphisms (SNPs) separating isolates to identify possible transmission pairs, together with epidemiological investigation. Tong et al, in common with other studies (*Young et al., 2012*; *Golubchik et al., 2013*; *Price et al., 2014*), expressed caution that high *S. aureus* diversity in the source individual would reduce the sensitivity of this approach. Nevertheless, it may be the best available method for reasons of cost or circumstance. In those cases, the SNP threshold must be carefully chosen. The value used in past work has been between 23 and 40 SNPs (*Price et al., 2014*; *Long et al., 2014*; *Azarian et al., 2015*; *Uhlemann et al., 2014*), but at the same time pairwise distances as large as 40 SNPs have been encountered within-host (*Golubchik et al., 2013*). As sampling within-host diversity provides an alternative, more sophisticated method to identify transmission pairs, the results of an analysis utilising it can be used to test different threshold values, in order to aid researchers who have access only to single genomes.

In this paper, we return to the study previously described by Tong et al. We use a considerably expanded version of that dataset, which adds more extensive within-host sampling. Our objectives were to reconstruct transmission between patients in the study using the new methodological insights mentioned above, to examine the bottleneck at MRSA transmission, to investigate the movement of the bacterium between body sites of individual hosts (that is, its phyloanatomy [*Salemi and Rife, 2016*]), and to compare our identification of transmission pairs with those obtained using an SNP-based approach.

# Results

## Patients and setting

Recruitment to the study was described previously (*Tong et al., 2015*). Briefly, consenting patients admitted to two ICUs (a paediatric unit and a general adult surgical unit) at Sunpasitthiprasong Hospital, Ubon Ratchathani, Thailand, during a period of three months in 2008 were recruited. MRSA screening was performed on admission and then twice weekly until discharge. Nasal swabs and fingertip cultures were also taken from ICU health care workers (HCWs) at three time points. Each screen consisted of swabbing the anterior nares, throat (or endotracheal suction tube if intubated), axilla, catheter urine if catheterised, and wounds if present (including pressure sores). Microbiological culture and bacterial identification have also been described previously (*Tong et al., 2015*). Up to ten colonies on primary culture plates were saved per sample. All colonies from nasal swab cultures were selected for sequencing, together with up to one colony from cultures from each other positive body site. For one subject (an adult ICU patient designated T126), additional colonies were collected, up to a maximum of 29 for some time points, also as described previously (*Tong et al., 2015*).

The complete set of subjects screened consisted of 169 adult patients, 98 child patients and 37 HCWs. To the dataset of Tong et al., which comprised up to three sequences from each subject for a total of 76, we added 923 more, for a total of 999. Of these, four were excluded as suspected contaminants, leaving 995, from 55 subjects (compared to 51 in the previous study). Five subjects were HCWs, 21 were surgical ICU patients, and 29 were paediatric ICU patients. The number of sequences per subject ranged from 1 to 239 (mean 18). In addition, 20 sequences (one per patient) for isolates originating from an earlier study in the same hospital (*Harris et al., 2010*) were included for the purposes of comparison. All isolates belonged to ST 239.

## Phylogeny reconstruction

The complete core chromosome sequence alignment was 3,043,210 base pairs in length. Bayesian phylogenetic analysis was conducted using ExaBayes 1.5 (*Aberer et al., 2014*). *Figure 1* displays the 50% majority-rule consensus tree, rooted using the TW20 strain (*Holden et al., 2010*) as an outgroup. Eight clades are highlighted: those designated 1 to 5 are the five previously identified by *Tong et al. (2015)* but much enlarged by additional data, whereas clades 6 to 8 all include some sequences that were isolated tips in the phylogeny in that paper but are now part of larger clades. Of the 20 sequences from the earlier Harris et al. study, 16 were part of these clades (but generally basal tips within them) whereas the remaining four were isolated.

## Identification of potential multiple colonisation events

The transmission process amongst the study subjects was investigated using *phyloscanner* v1.4.2 (*Wymant et al., 2017*). This tool, intended for the analysis of large genetic datasets of within- and between- host pathogen data, reconstructs the relative positions of hosts in the chain of transmission.

This reconstruction was performed with an awareness of the possibility that subjects may experience multiple independent infection or colonisation events, in which case each such event should be treated as a separate entity when investigating transmission. An initial *phyloscanner* investigation suggested that sequences from five subjects (designated T035, T099, T159, T271 and T327) formed two phylogenetic clades where the median patristic distance between clade MRCAs (across the ExaBayes posterior) was greater than 100 substitutions, sufficiently diverse that they were unlikely to be the product of single-strain colonisation events. The sequences from each of these were separated into two groups and each was subsequently treated independently. We will refer to links in the transmission chain as 'colonisations'; each isolate belongs to a single colonisation. Numerical codes prefixed with a 'T' are used for study subjects and codes with 'C' for colonisations. In most cases the numbers after this prefix match; for example C009 is the sole colonisation of subject T009. For the five multiply-colonised subjects, a lowercase letter is used to differentiate them, for example C035a and C035b are the two colonisations of subject T035.

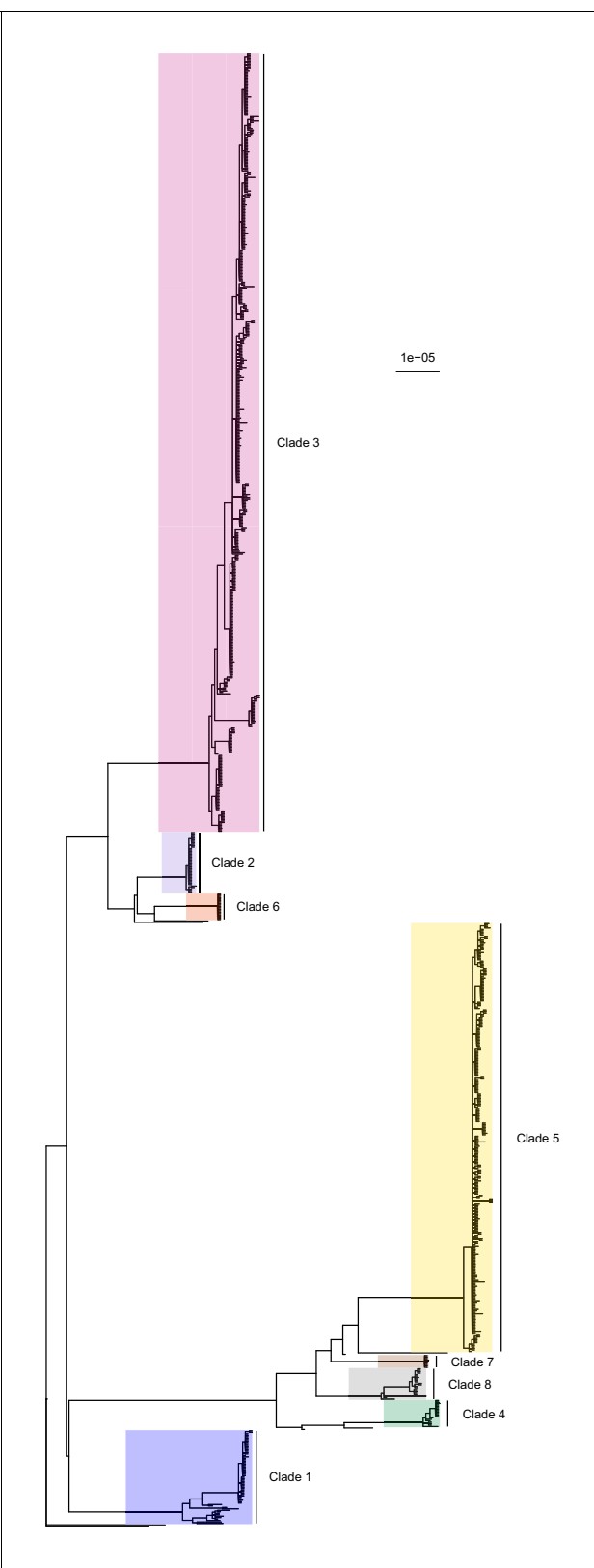

**Figure 1.** 50% majority-rule consensus tree of the posterior distribution of ExaBayes phylogenies. Branch lengths are in substitutions per site. Eight clades are highlighted. Clades 1 to 5 correspond to those identified by *Tong et al. (2015)*, while the additional three are newly designated.

The online version of this article includes the following source data for figure 1:

**Source data 1.** ExaBayes consensus tree in Newick format.

## Trace colonisations

Of 60 identified colonisations from the 55 subjects, cultures for 33 came only from a single positive swab. Those for the remaining 27 were obtained either on at least two time points, from at least two body sites, or both. The median number of available sequences per colonisation was 1, but this rises to 14.5 with these singletons excluded. In 25 of the 33 single-swab colonisations, previous and/or subsequent examinations of the same patient did not identify MRSA belonging to the same colonisation, while in eight only one examination was performed. In what follows we refer to these 33 as trace colonisations.

When compared to those from multiple isolations, sequences from trace colonisations showed greater similarity to those from isolates collected from other study subjects at an earlier time. *Figure 2* displays the distribution of the median number of SNPs separating the isolates in each colonisation from the most closely related isolate sampled on an earlier date. There was strong evidence supporting a difference in the distribution of these distances between trace and non-trace colonisations (Mann-Whitney *U* test p=0.018), suggesting that the nature of trace isolates could be fundamentally different to that of typical isolates from a non-trace colonisation, rather than being simply the product of colonisations of a similar nature subject to sparser sampling. A similar pattern was observed when the median SNP distance was calculated from only those isolates in a non-trace colonisation that were acquired at the time of the first positive swab (*Figure 2—figure supplement 1*) although in this case statistical support was lacking (p=0.119). For six trace colonisations (C104, C105, C223, C225, C270, and C271b), our subsequent *phyloscanner* analysis inferred an infector from amongst the patients already or previously present in the hospital at the time that the sample in question was taken. For the six patients corresponding to these colonisations, the positive swab was acquired on the day of or day after ICU admission, and within three days of hospital admission. Such a short time from hospital admission to a positive swab with a sample very similar to those already existing in the hospital was not observed for any non-trace isolates. Four of the six swabbed negative for MRSA on a subsequent occasion. These observations suggest that at least some of the trace category were of a different nature to colonisations providing multiple isolates, and that they potentially represent incidental, transient exposure of patients to strains in the overall hospital environment. The possibility of sample contamination also cannot be excluded in any individual example.

Our main interest is in the transmission of established, rather than transient, colonisations, as these represent the bacterial reservoir from which further colonisations will be derived, meaning they should be prioritised when designing interventions for infection control. The only strong evidence available to us that a given sample is not the result of transient colonisation is the acquisition of multiple samples of the same strain. The confirmation of colonisation with multiple swabs also greatly reduces the possibility of contamination. As a result, we exercise caution and omit the trace category from consideration for the bulk of the main text of this paper. Versions with them included are presented as figure supplements.

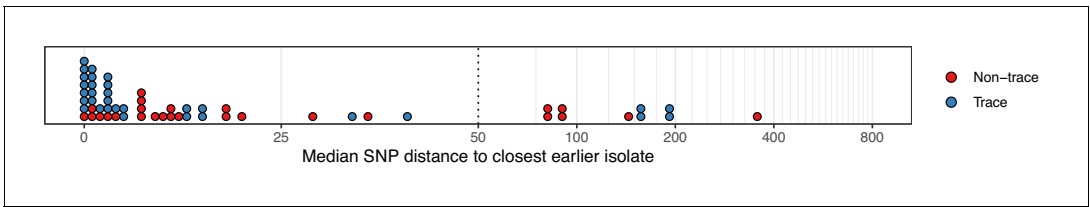

**Figure 2.** The median number of SNPs separating sequences from each colonisation from the most similar sequence from an isolate acquired before the date of the first positive sample from that colonisation. Blue dots are colonisations isolated from a single swab only (trace colonisations), while red were acquired from colonisations where multiple swabs were isolated. For three colonisations (two trace) the first collection date was the commencement of the study and hence there was no such earlier isolate. The x-axis transfers to a log scale on the right of the dotted line.

The online version of this article includes the following source data and figure supplement(s) for figure 2:

**Source data 1.** Source data and R script for creation of *Figure 2* and *Figure 2—figure supplement 1*.
**Figure supplement 1.** The median number of SNPs separating sequences from each colonisation, restricted to those obtained at the time of the first positive swab, from the most similar sequence from an isolate acquired before the date of the first positive sample from that colonisation.

## Reconstructed transmission events

*Phyloscanner* leverages the signal that transmission leaves on the phylogenetic topology (*Romero-Severson et al., 2016*) in order to reconstruct the direction of transmission between colonised individuals. It works by performing ancestral state reconstruction on a phylogeny using maximum parsimony, using the set of colonisations as states. It identifies phylogenetic relationships between colonisations by examining the arrangement of *subgraphs*: contiguous blocks of phylogenetic nodes which all have the same reconstructed state. Pairs of colonisations that are closely related in the transmission chain are identified by proximity of their subgraphs and the direction of transmission between them inferred from their topological arrangement, as shown in *Figure 3*. We performed this procedure on the consensus phylogeny shown in *Figure 1*, but also, to allow for phylogenetic uncertainty, each of 100 random trees from the ExaBayes posterior.

*Figure 4* superimposes the results of the reconstruction using the consensus phylogeny onto the timeline of patient stays in the hospital. The timings of positive and negative screening events are indicated by circles and crosses respectively. Horizontal lines represent hospital and ICU stays and are coloured by population (hospital ward or HCW). Reconstructed transmission events are indicated by grey arrows, appearing at a time indicating the upper bound for the date at which the transmission could have occurred (the earliest time of sampling amongst the tips in the recipient subgraph). Multiple arrows appearing between the same two subjects (for example between C012 and C159b) suggest the transmission of multiple lineages, either simultaneously or over a more extended period of time. All such events reconstructed here are within-ward, although four between-ward events were reconstructed with trace colonisations as recipients (see *Figure 4—figure supplement 1*). The great majority of events were consistent with the timeline of hospital and ICU stays, allowing that subjects may act as infectors subsequent to their departure from the premises due to environmental contamination or an unsampled intermediary carrier. Two exceptions are the descent of C271a from C327a when subject T271 left the hospital prior to subject T327's arrival, and descent of C009 from C159a prior to the arrival of T159.

## Epidemiologically plausible transmissions have high posterior support

To see if these apparently impossible reconstructed events were phylogenetically well-supported, we investigated whether the pattern persisted when the analysis was performed on the 100 sampled posterior trees. Specifically, for every subgraph of each host in each tree, we identified the last sampled colonisation in the transmission chain (if any), and checked whether that transmission was consistent with the timings of hospital stays and sample collection dates. *Figure 5* shows the results. The transmission from C327a to C271a had posterior support of 1, but this would disappear entirely

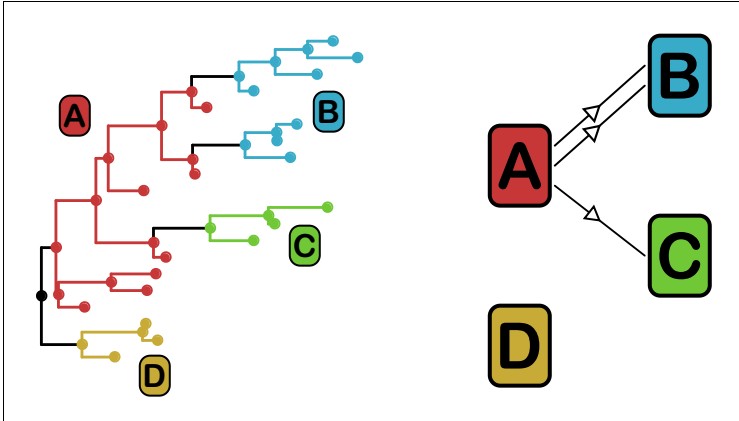

**Figure 3.** *phyloscanner* identifies transmission pairs by ancestral state reconstruction of hosts (left) and subsequent classification of the topological relationships between the subgraphs reconstructed to each host (right). In this example the hosts are designated (**A to D**). Here host A is inferred to be the infector of hosts (**B and C**). The transmission from (**A to C**) was of only a single pathogen lineage, while that from A to B was of two, with the result that host B has two subgraphs. The subgraph from host D forms a sibling clade to the rest of the phylogeny and, as a result, no inference is made about transmission.

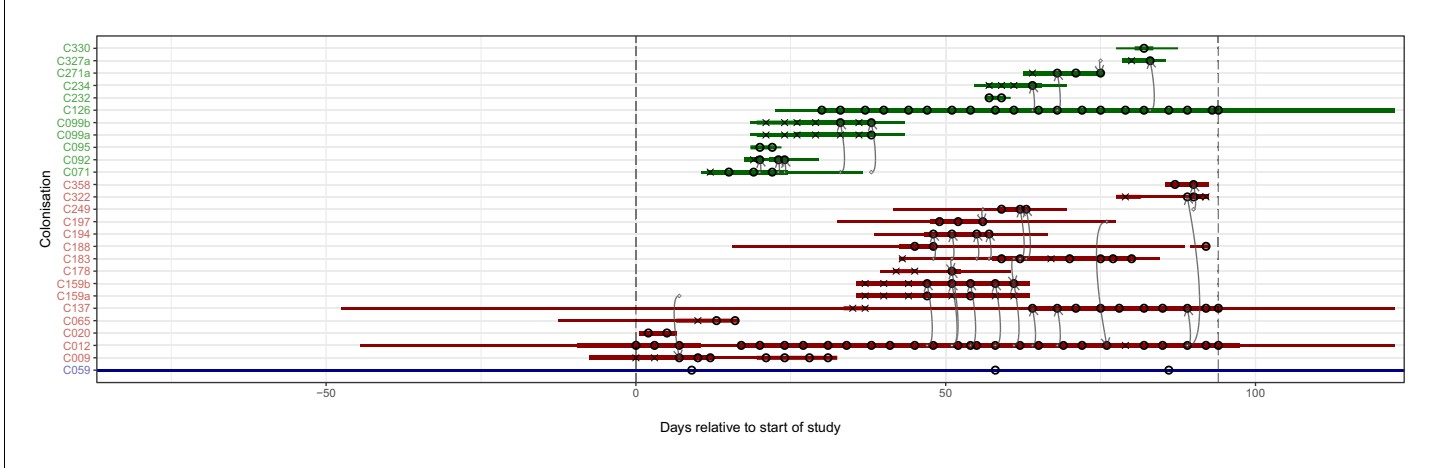

**Figure 4.** The reconstruction of the transmission process using the consensus tree overlaid on a timeline of hospital and ICU stays and sampling events. Each row represents a colonisation, with thin lines representing the colonised subject's presence in the hospital and thick lines their presence as a patient in an ICU. Colours of the lines and the *y*-axis labels indicate surgical ICU patients (green), paediatric ICU patients (red) and HCWs (blue). Crosses represent times of screens that were negative for MRSA, while circles those that returned positive swabs and sequenced isolates. The grey arrows represent reconstructed transmission events. These appear when at least one subgraph from the recipient is descended from an adjacent subgraph from the infector. Such a transmission may also involve unsampled intermediaries or the environment. The timings of these arrows represent the upper bound for the time at which they could have occurred rather than an exact estimate. The dotted vertical lines demarcate the period of sampling.

The online version of this article includes the following source data and figure supplement(s) for figure 4:

**Source data 1.** Source data and R script for creation of *Figure 4* and *Figure 4—figure supplement 1*.
**Figure supplement 1.** The reconstruction of the transmission process using the consensus tree overlaid on a timeline of hospital and ICU stays and sampling events, with trace colonisations included.

---

if a single tip from C271a was removed. Subgraphs from four other colonisations (and an additional six trace colonisations, see *Figure 5—figure supplement 1*) were the result of impossible infection events that had non-negligible support (in the range from 0.28 to 0.76). Two of these involved patient T159 as the source, and the remaining two involved the closely related triplet of C071, C092, and C099b.

We performed a separate analysis where the set of tips from each subject was randomly downsampled to a maximum of five. This resulted in considerably more impossible reconstructed events (*Figure 5—figure supplement 2*).

## Transmission clusters

*Figure 6* presents the *phyloscanner* host relationship diagram, displaying the division of the 27 colonisations into 12 clusters of closely-related infections. (Clusters here indicate groups of colonisations linked by any number of reconstructed transmission events, and no genetic distance threshold was applied.) Links between colonisations are made up of three segments, whose colour represents the frequency of a given topological relationship between two colonisations in the ExaBayes posterior. The outer two, which have directional arrows, represent transmission in the direction of the arrow, while the central segment represents the 'complex' ancestral relationship (*Wymant et al., 2017*), which suggests a close genetic relationship for which the direction of transmission is unclear. Links are also labelled with the proportion of posterior trees featuring any one of these three relationships. To avoid excessive clutter in the figure, they are only displayed when this number is 0.5 or greater. Two of the large clusters revolve around the long-staying patients T012 in the paediatric ICU and T126 in the general ICU. 8 of the 12 clusters are singletons, representing colonisations not closely related to any other sample in this study. *Figure 7* gives the equivalent diagram from an analysis where each subject's isolates were further subdivided by the body site of origin; links with coloured backgrounds connect colonisations from the same patient. The same figures with trace colonisations included can be seen in *Figure 6—figure supplement 1* and *Figure 7—figure supplement 1*. Separating colonisations by body site can clarify relationships, most strikingly for C249,

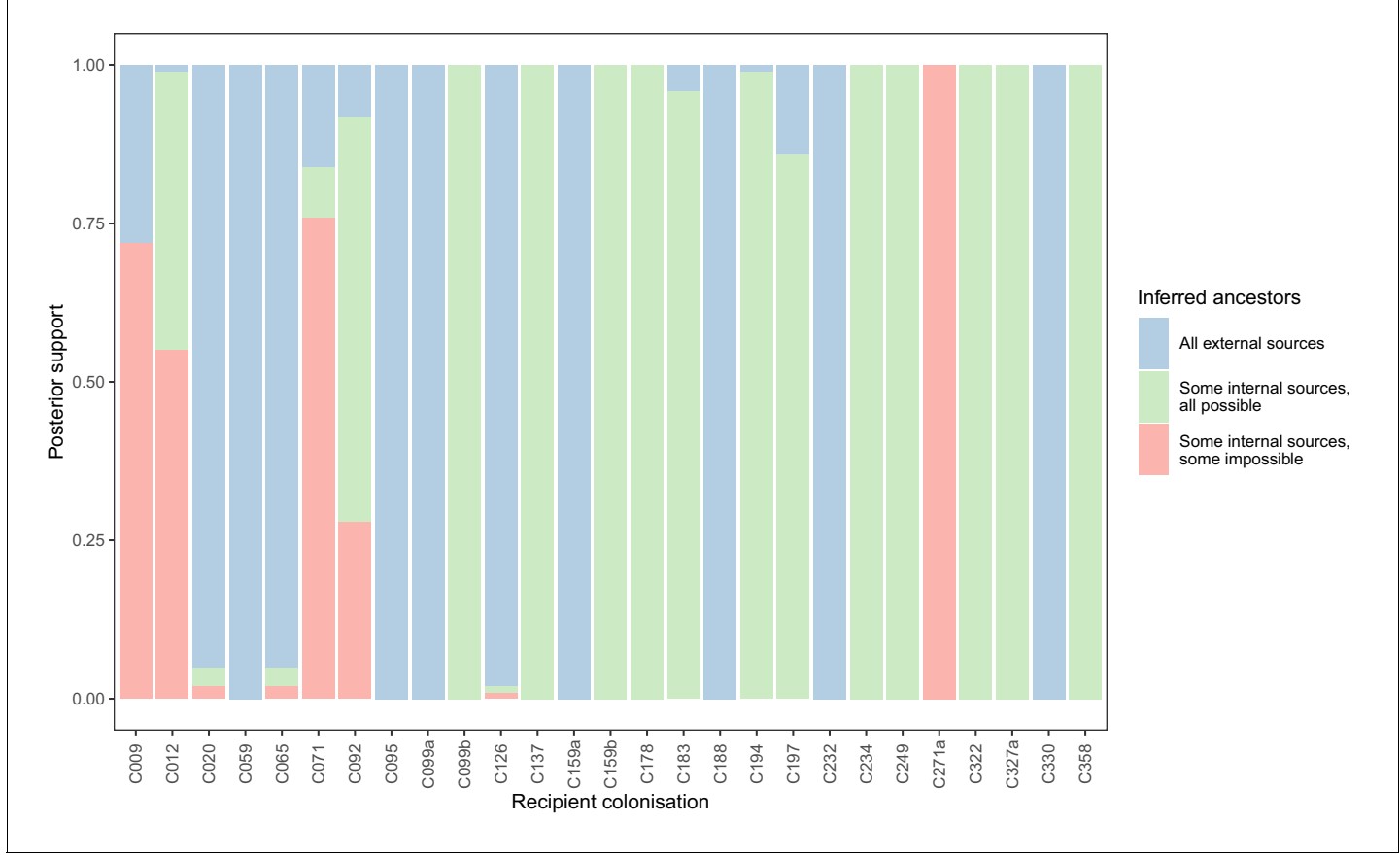

**Figure 5.** Concordance of inferred infectors for each colonisation with recorded timings of hospital stays and sampling dates. Each bar represents a colonisation, and the colours represent the proportions of the posterior set of trees where the transmission chain prior to that host involves no sampled subjects (blue), involves one or more sampled subjects all of which are possible given known timings of entry and departure to the hospital and sampling of isolates (green) and at least one sampled subject where the timings are in conflict, with the infector entering the hospital after isolates from the recipient were acquired (red).

The online version of this article includes the following source data and figure supplement(s) for figure 5:

**Source data 1.** Source data and R scripts for creation of *Figure 5*, *Figure 5—figure supplement 1* and *Figure 5—figure supplement 2*.
**Figure supplement 1.** Concordance of inferred infectors for each colonisation with recorded timings of hospital stays and sampling dates, with trace colonisations included.
**Figure supplement 2.** Concordance of inferred infectors for each colonisation with recorded timings of hospital stays and sampling dates, from a secondary *phyloscanner* analysis in which the tip set from each subject was randomly downsampled until a maximum of five remained.

whose close relationships with a large number of other colonisations, with no clear directionality, in *Figure 6* is resolved into separate origins for the samples with an axillary origin, which show support for being transmitted from either C183 or C194, and the remainder, which are likely to have come from C197 or C358. Note that these diagrams are representations of the pairwise relationships between colonisations and do not attempt to resolve these into a single transmission history.

## Between-subject transmission bottlenecks are of very variable size

We investigated the size of the bottleneck at transmission for six pairs of colonisations in detail. (By 'bottleneck' we refer to the total number of genetic lineages passed from one colonisation to another; in this study we cannot differentiate between multiple lineage transmission at a single time point and the transmission of multiple single lineages at different points.) All six had a reconstructed transition in one or both directions in 95% of posterior trees, and, in the subgraphs involved in those transmissions, a posterior median of at least five tips in those belonging to the recipient. The latter condition ensured that a reasonable number of sequences were sampled from the recipient, to

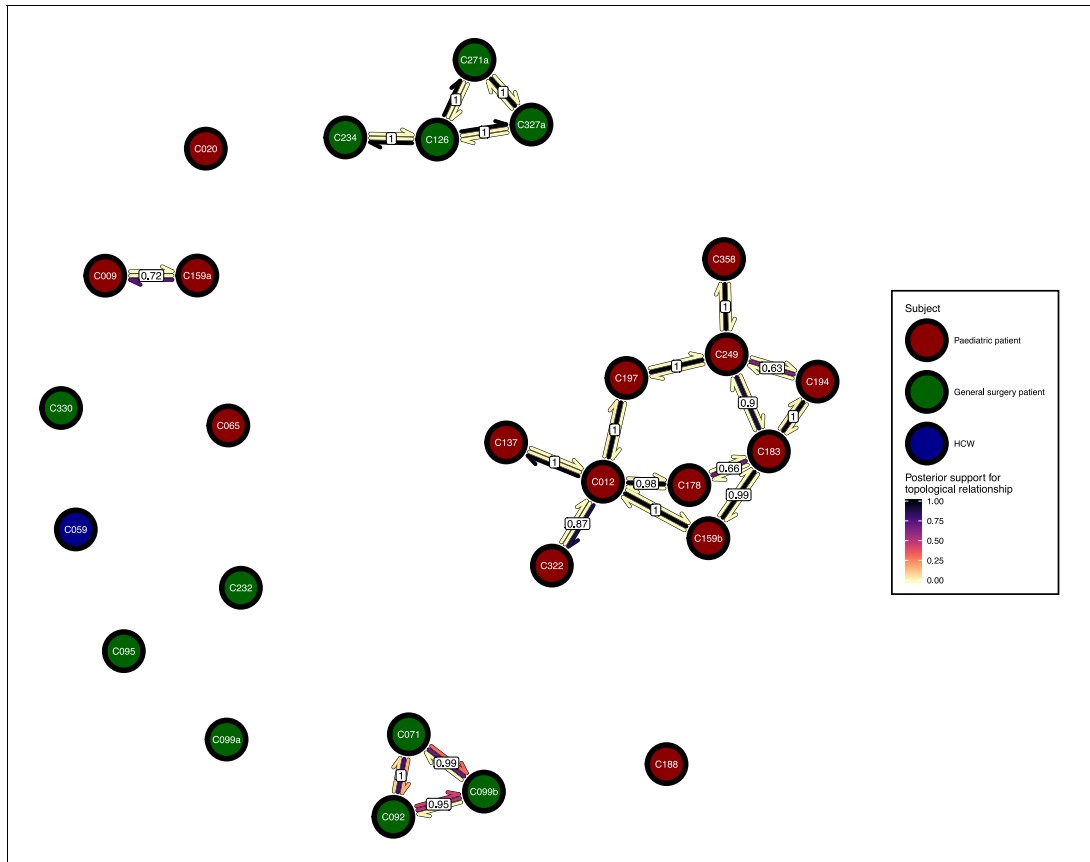

**Figure 6.** The *phyloscanner* host relationship diagram. Each node represents all the sequences for one colonisation. Node fill colours designate patients in the two hospital ICUs and the HCWs. Edges appear where colonisations share a relationship with posterior support of at least 0.5 and consist of three elements: arrows representing transmission in either direction and a central line segment representing the 'complex' topological relationship, which is indicative of transmission but the direction is ambiguous. Each of these is coloured according to the proportion of posterior trees showing the corresponding relationship. Edges are also labelled with the overall posterior support for any topology suggesting transmission. The online version of this article includes the following source data and figure supplement(s) for figure 6:

**Source data 1.** Source data and R script for creation of *Figure 6* and *Figure 6—figure supplement 1*.
**Figure supplement 1.** The *phyloscanner* host relationship diagram, with trace colonisations included.

prevent the inference of narrow bottlenecks due simply to low sequence counts. These pairs are summarised in *Table 1*. The transition counts represent a lower limit on the number of lineages transmitted from one subject to another that are necessary to explain the phylogeny.

Two very long-stay ICU patients, T012 and T126, both appeared to transmit to other patients through bottlenecks of quite variable strength. C012 transmitted to C137 through a relatively narrow bottleneck and C159b through an extremely wide one, and the corresponding patients' ICU stays each overlapped with T012 for at least four weeks, with both occupying a neighbouring bed to T012 for some (but not all) of that time. C126 transmitted a single lineage to C271a in every posterior tree, but showed a wider bottleneck in transmitting to both C234 and C327a. Subject T271 occupied the adjacent bed to subject T126 for 10 days but neither T234 nor T327 ever did.

Finally, C183 and C194 had such similar sequences that the reconstruction suggested back-and-forth movement of lineages. However, this may be due to poor phylogenetic resolution rather than the literal truth, especially as subjects T183 and T194 were never simultaneously present in the ICU. A third patient in that ICU, T178, also swabbed positive with a strain identical to some found in both T183 and T194 and one bed was at different times occupied by all three subjects, while a fourth, T249, was also colonised with a closely-related strain and occupied a bed which was later briefly used by T183. We lack the resolution necessary to definitively determine the order of events, but it

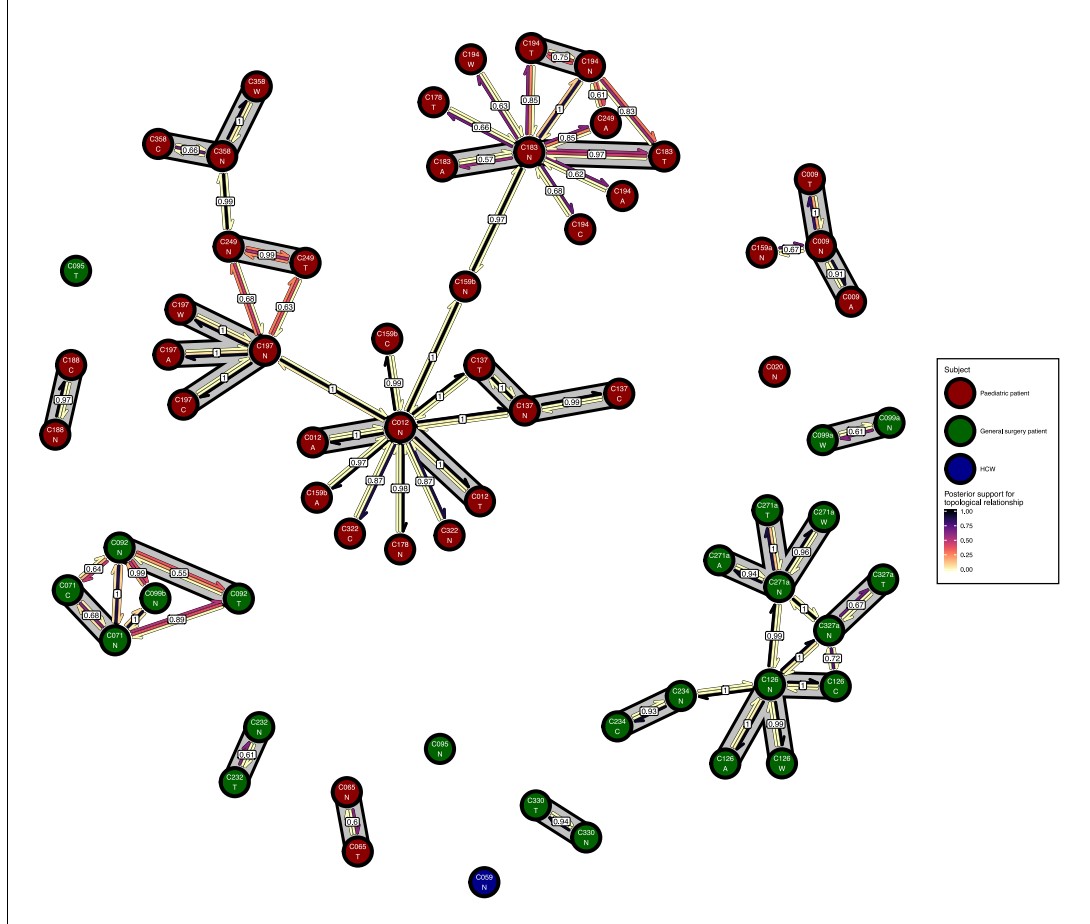

**Figure 7.** The *phyloscanner* host relationship diagram for a separate analysis where samples taken from distinct body sites on the same subject were treated as separate 'hosts'. Each node represents all the sequences for one colonisation of one body site. Node fill colours designate patients in the two hospital ICUs and the HCWs. Edges appear where colonisations share a relationship with posterior support of at least 0.5 and consist of three elements: arrows representing transmission in either direction and a central line segment representing the 'complex' topological relationship, which is indicative of transmission but the direction is ambiguous. Each of these is coloured according to the proportion of posterior trees showing the corresponding relationship. Edges are also labelled with the overall posterior support for any topology suggesting transmission, and edges connecting colonisations from sites from the same subject have a grey background. Nodes are annotated with colonisation IDs and a code for body site: A = axilla, C = endotracheal suction, N = nose, T = throat, W = wound.

The online version of this article includes the following source data and figure supplement(s) for figure 7:

**Source data 1.** Source data and R script for creation of *Figure 7* and *Figure 7—figure supplement 1*.
**Figure supplement 1.** The *phyloscanner* host relationship diagram for the body site analysis, with trace colonisations included.

seems most likely that, if there was no back-and-forth transmission, the transmission bottleneck was wide.

*Table 1* also presents figures for the nucleotide diversity in the recipient population for these six events. This shows no obvious correlation with bottleneck size, indicating that the diversity in a sample may have be transmitted, but may also have arisen within-host.

## The majority of subjects showed no evidence of phylogeny/body site correlation

23 study subjects had tested positive for MRSA colonising at least two different body sites. In every case, one site was the anterior nares. Two subjects (T178 and T322) provided only a single sequence from each of two sites, which rendered them unsuitable for further investigation as only one phylogenetic topology is possible under that circumstance. To investigate phylogeny-site associations in the remaining 21 subjects, we ran ExaBayes on the sequences from each separately and analysed

**Table 1.** Six pairs of colonisations for whom transmission was reconstructed with a posterior score of at least 0.95 and, for those reconstructed transmissions, a posterior median of at least five tips were in the recipient subgraphs.

Each row gives the posterior median and the limits of the 95% highest posterior density (HPD) interval for the number of reconstructed transmissions between the subjects, in either direction. The posterior median number of tips in the recipient subgraphs, and the nucleotide diversity amongst those tips, are also given. In five cases the inferred direction of transmission is clear, but for C183 and C194 it is not.

| Colonisation A | Colonisation B | Number of A to B transitions 95 % HPD | | | Tips in descendant B subgraphs (median) | Nucleotide diversity transmitted to B (median) | Number of B to A transitions 95% HPD | | | Tips in descendant A subgraphs (median) | Nucleotide diversity transmitted to A (median) | Direction |
|---|---|---|---|---|---|---|---|---|---|---|---|---|
| | | Median | Lower | Upper | | | Median | Lower | Upper | | | |
| C012 | C137 | 3 | 2 | 3 | 79 | 1.46E-06 | 0 | 0 | 0 | 0 | NA | A to B |
| C012 | C159b | 34 | 25 | 38 | 39 | 2.20E-06 | 1 | 0 | 6 | 1 | 1.40E-06 | A to B |
| C126 | C234 | 3 | 3 | 3 | 10 | 1.92E-06 | 0 | 0 | 0 | 0 | NA | A to B |
| C126 | C271a | 1 | 1 | 1 | 21 | 3.11E-07 | 0 | 0 | 0 | 0 | NA | A to B |
| C126 | C327a | 5 | 4 | 5 | 9 | 2.68E-06 | 0 | 0 | 0 | 0 | NA | A to B |
| C183 | C194 | 15 | 1 | 23 | 31 | 3.41E-07 | 5 | 0 | 19 | 12 | 7.08E-07 | Unclear |

the results using BaTS (*Parker et al., 2008*), to identify deviations from the null hypothesis of no phylogeny-site association according to the association index (AI), parsimony score (PS) and the size of largest monophyletic clade (MC) for each trait.

Deviation from the null in the AI statistic indicates the presence of phylogenetic nodes whose descendants are more frequently associated with some sites than would be expected by chance. This is the most sensitive of the three statistics to the existence of an association, but such an association does not imply the existence of the distinct monophyletic clades that would be expected if sites were separately colonised by different lineages, or if a single lineage from one site was transmitted to another. For the PS and the MC statistics, deviation from the null is much more suggestive of site-specific monophyletic clades. However, the PS statistic is invariant if all but one tip in the tree comes from the same site, as is the MC statistic for any site with just a single tip, and hence these statistics are useless in this scenario. If some sites have only one tip and the AI statistic shows significant deviation from the null, then further examination is required to check if, for example, the tip in question is basal. See *Supplementary file 1* for some example phylogenies and statistic values.

*Table 2* summarises the results of this analysis. For 11 of 21 subjects there was no evidence of any association between a tip's position in the phylogeny and the site the corresponding sequence was acquired from. For another six, the only statistic to show deviation from the null at the p=0.05 level was the AI. The lack of significance for the PS suggested that no site-specific monophyletic clades were present, but this cannot identify divergent singleton tips. Examination of the ExaBayes consensus trees by eye revealed singletons for only one patient, T271. In this case the divergence (431.2 SNPs) of a single isolate from an endotracheal suction tube from the remaining isolates was so large that it had already been taken into account by the separation of the patient's sequences into colonisations C271a and C271b; the trace colonisation C271b is the single endotracheal sequence.

The remaining four subjects are T009, T126, T232 and T249, where the ExaBayes topologies were analysed for the signals of separate colonisation events and single lineage transmission from one site to another. Consensus trees are displayed in *Figure 8*. Posterior support for either signal in the tree sets for T009 and T126 was negligible or absent (less than 0.05). For T249, the axial samples formed a separate clade with posterior support of 1, but trees in which these were nested within the remaining diversity were absent from the posterior; it can be seen from *Figure 7* that these probably originated with a different subject from the rest. For T232 there was considerable support for both nasal and throat samples forming clades (0.712 and 0.513) respectively, and also for the diversity of each being nested in that of the other (0.514 and 0.482).

Ultimately, only three of the 21 subjects (T232, T249 and T271) had phylogenies consistent with separate colonisation events affecting different body sites or groups of sites. Only T249

**Table 2.** Results of the investigation of body site/phylogeny associations.

Each row corresponds to a single BATS analysis on a posterior set of phylogenies consisting just of the sequences from that subject. The p-values are given for the association index (AI), parsimony score (PS) and largest monophyletic clade (MC) size for all body sites present in the dataset. Values with an asterisk (*) correspond to statistics whose values are identical under both hypotheses due to singleton sequences from some sites.

| Subject | Number of Sequences | p-value | | | | | | |
| | | AI | PS | MC | | | | |
| | | | | Axilla | Nose | Throat | Trachea | Wound |
|---|---|---|---|---|---|---|---|---|
| T009 | 31 | 0.026 | 1 | 1* | 0.031 | 1 | | |
| T012 | 223 | 0.24 | 0.18 | 1 | 1 | 0.14 | | |
| T065 | 11 | <0.001 | 1* | | 0.29 | | 1* | |
| T071 | 13 | 0.16 | 1 | | 0.16 | | 1 | |
| T092 | 12 | 0.035 | 1 | | 1 | 1 | | |
| T095 | 11 | 0.099 | 1* | | 0.1 | 1* | | |
| T099 | 19 | 1 | 1* | | 0.16 | | | 1* |
| T126 | 239 | <0.001 | <0.001 | 0.004 | 0.1 | | 0.004 | 1* |
| T137 | 79 | 0.004 | 1 | | 0.12 | 1 | 1* | |
| T159 | 46 | 0.7 | 1 | 1* | 1 | | 1 | |
| T183 | 48 | 0.04 | 1 | 1* | 0.22 | 1 | | |
| T188 | 19 | 1 | 1* | | 1 | | 1* | |
| T194 | 32 | 0.13 | 1 | 1* | 1 | 0.37 | 1* | 1* |
| T197 | 35 | 0.74 | 1 | 1 | 1 | | 1 | 1 |
| T232 | 11 | <0.001 | <0.001 | | 0.001 | 0.001 | | |
| T234 | 10 | 1 | 1* | | 1 | | 1* | |
| T249 | 14 | 0.002 | <0.001 | 0.017 | 0.016 | 1 | | |
| T271 | 23 | 0.043 | 1 | 1* | 0.49 | 1 | 1* | 1* |
| T327 | 10 | 1 | 1* | | 0.51 | 1* | | |
| T330 | 8 | 1 | 1* | | 0.26 | 1* | | |
| T358 | 22 | 0.004 | 1 | | 0.06 | | 1 | |

exhibited one nested clade from a single site, which would be indicative of a single transmission from another site through a narrow bottleneck, but statistical support for this remains limited.

## Identification of transmission pairs from single genomes

Previous studies have not had access to such rich within-host sampling, and have frequently identified transmission pairs using SNP distance between individual isolates from hosts. As our data admits a more powerful approach based on the topology, we compared our results to those from a hypothetical study where only a single sequence is available from each subject. We used phyloscanner as a gold standard for identification of transmission pairs, and pairs of subjects that were definitively not transmission pairs. For the latter we used posterior support of less than 0.05 for having adjacent subgraphs and a topological relationship implying ancestry. We used two standards for positively identifying pairs. The first was in inverse of the previous, that is support greater than 0.95 for having adjacent subgraphs and a topological relationship implying ancestry. This version, however, has some problems as a gold standard as it does not attempt to eliminate the possibility of unsampled intermediaries in the transmission chain. For a second version where these are much less likely, we used support greater than 0.95 for having adjacent subgraphs and a topological relationship that implied the transmission of multiple lineages. This is the equivalent of the 'PP' topology of Romero-Severson et al. (*Romero-Severson et al., 2016*) which was shown to be highly suggestive of direct transmission. (Intuitively, for a missing intermediary to be present in this scenario, multiple

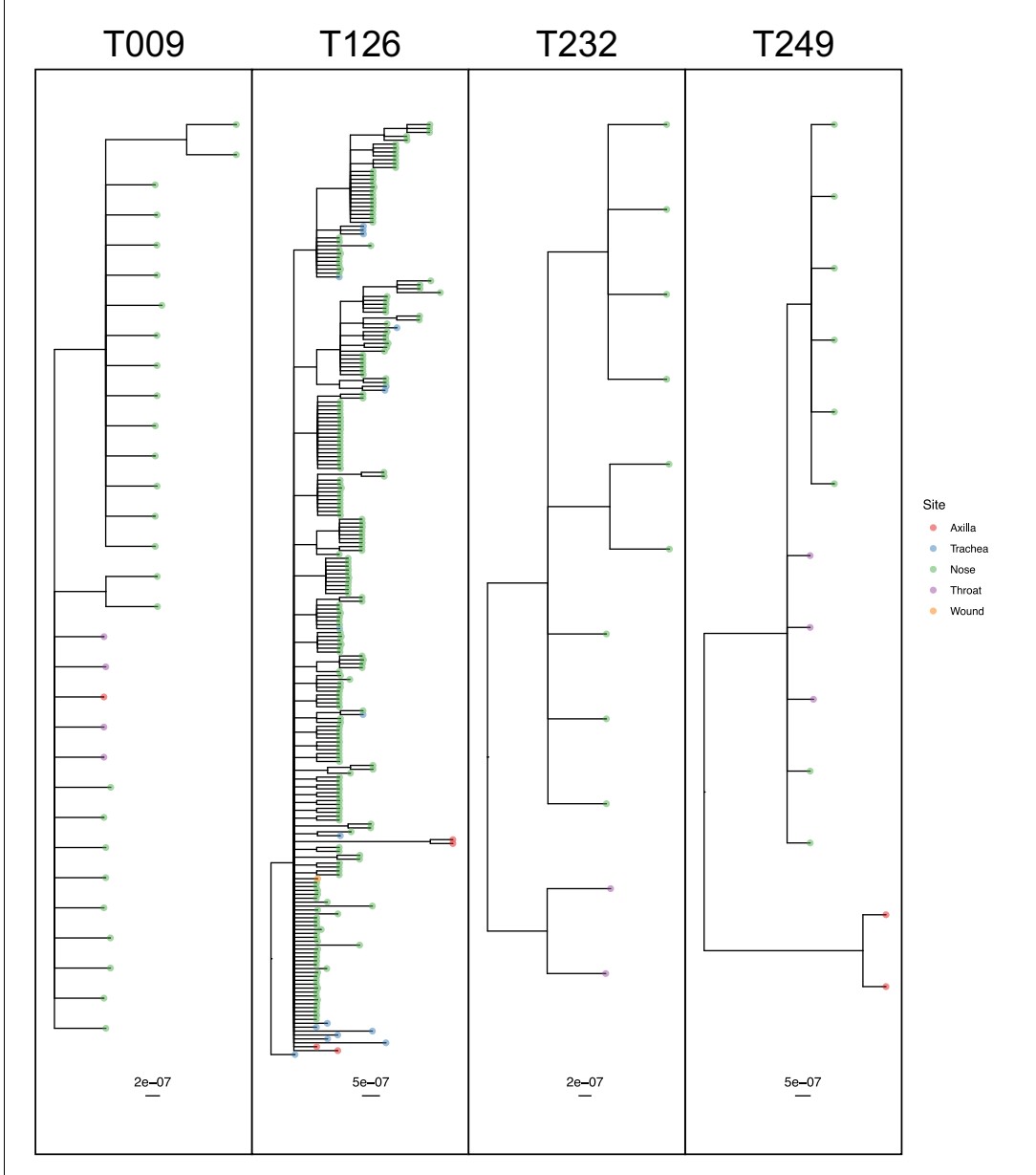

**Figure 8.** 50% majority-rule consensus phylogenies for the ExaBayes phylogenetic analyses of the sequences from patients T009, T126, T232 and T249. Tips are coloured by body site of origin. Branch lengths are in substitutions per site. Trees were rooted using the TW20 outgroup (not shown). The online version of this article includes the following source data for figure 8:

**Source data 1.** ExaBayes consensus trees for all subjects analysed in the phyloanatomy analysis.

lineages would need to be transmitted at least twice, which is a rarer event than repeated transmission of at least one lineage.) This classification, however, means that subjects from whom only trace colonisations were identified have to be excluded, as it is impossible to reconstruct multiple lineage transmission if one subject contributes only a single tip to the phylogeny (or both do). As this is an analysis from which there is little reason to remove trace colonisations, which would not be identifiable in our hypothetical single-sample study, we present both versions.

For each subject, we used the samples taken at the first positive swab as potential samples from a single-sequence study. For each pair of these acquired from different subjects, we calculated the raw SNP distance and used these figures to evaluate the performance of SNP distance as means to determine transmission pairs and members of clusters, using the above two measures as gold

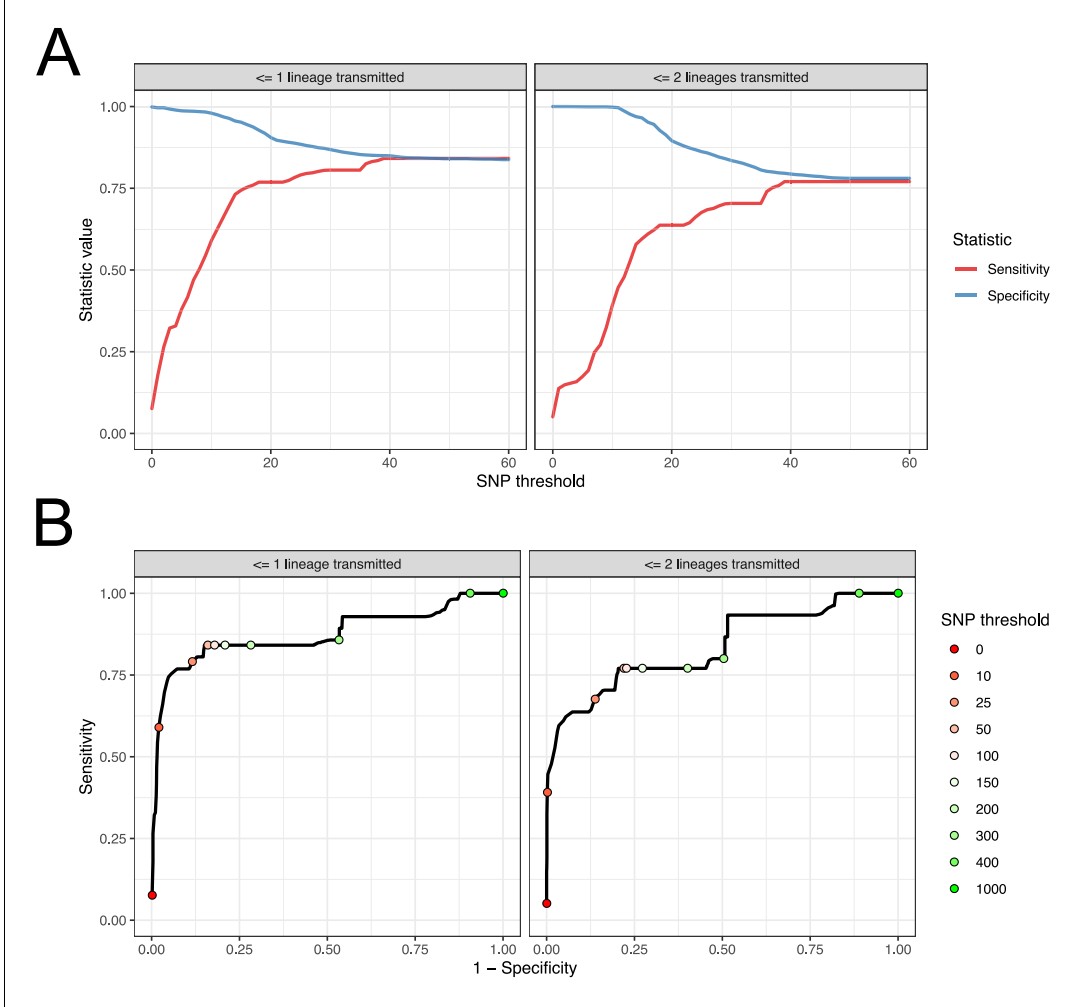

**Figure 9.** Performance of SNP distance as a method for identifying transmission pairs. (**A**) Plots of the sensitivity and specificity of using the number of SNPs to identify transmission pairs, for different distance thresholds. (**B**) ROC curves plotting true positive rate (sensitivity) against false positive rate (1-specificity) for different thresholds. The curve is annotated with selected threshold values. The gold standard for identifying transmission pairs in the version on the left is a topological relationship suggesting at least one transmitted lineage, while on the right at least two are required, a criterion which will occur much less often if there is a missing intermediary in transmission.

The online version of this article includes the following source data and figure supplement(s) for figure 9:

**Source data 1.** Source data and R script for creation of *Figure 9* and *Figure 9—figure supplement 1*.
**Figure supplement 1.** Plots of the sensitivity and specificity of using the number of SNPs to identify transmission pairs, for different distance thresholds.

standards. The results can be seen in *Figure 9*. Subfigure A shows how the sensitivity and specificity vary with the threshold. To prevent bias in these results due to variable sample counts per subject, these figures are weighted by the probability of (uniformly) selecting a given pair of sequences from all those available from the same pair of subjects. (The maximum threshold shown in this figure is 60 SNPs, but see *Figure 9—figure supplement 1* for the range up to 500 SNPs.)

For both gold standards, SNP distance has generally good specificity for reasonable thresholds, staying above 0.75 for less than 140 SNPs. Sensitivity reaches 0.75 at a threshold of 16 SNPs for the version which is indifferent to unsampled intermediaries, but this requires 37 SNPs for the strict version.

Subfigure 9B shows receiver operating characteristic (ROC) curves for the same investigation. The area under the curve statistic is 0.872 for the first gold standard and 0.836 for the second, while the

value of the threshold which minimises the distance to the upper-right corner of the plot (if sensitivities and specificities are equally weighted) is 39 SNPs for both analyses.

## Discussion

In this study, we used a large genetic dataset including within-host diversity to reconstruct MRSA transmission in a hospital setting, utilising new methods that investigate the phylogenetic topology. We found great variation in the size of the transmission bottleneck between study subjects, and that in the majority of cases there was no suggestion of an association between the phylogenetic placement of an isolate and the body site that it was acquired from. Finally, we determined SNP thresholds for identifying transmission pairs with high sensitivity and specificity.

33 out of 60 colonisations consisted of a single swab (trace colonisations). If the nature of these was precisely the same as the multiply-isolated colonisations and they were merely sampled less often, one would expect to see no difference in the distribution of genetic distances to isolates already identified in the hospital between the two groups. We did not find this to be the case. One explanation for this would be transient carriage, due to incidental exposure of patients to bacterial lineages already well established in the hospital environment. Indeed, thirteen trace colonisations came from patients who underwent a subsequent screen with negative results (and five underwent several such screens, which never occurred at all for non-trace colonisations). Previous studies have found transient carriage to be quite common in both patients (*Bradley et al., 1991*) and HCWs (*Cookson et al., 1989*). However, it remains impossible to rule out the possibility that any individual example was the result of a contamination event leading to the source individual being wrongly identified, especially given the very short times from hospital admission to a positive swab that are implied for six subjects. This leads us to recommend that in future phylogenetic studies, positive swab results should be verified by undertaking multiple sampling wherever possible.

The concordance of the order of colonisation given by the phylogenetic topology and the known dates of hospital stays was generally good; with the exception of one case (C271a) which plausibly involved contamination, those transmissions that contradicted the reported timings of stays and MRSA screens did not achieve posterior support above 0.76 and a threshold of 0.95 would comfortably eliminate them.

The role of variable sample counts per colonisation in an analysis of this sort is complex. The relative number of tips derived from one colonisation is information that the parsimony reconstruction will use in determining the source and recipient in a putative transmission pair; the colonisation with more tips is more likely to be reconstructed as the source in situations of phylogenetic uncertainty. While this may appear at first glance to be a bias, we see here (*Figure 5—figure supplement 2*) that removing the effect by equalising counts leads to a much poorer reconstruction. This is because sample count is associated with a variable, ICU stay length, which is itself associated with source status. Adjusting for it is hence adjusting for a variable on the causal pathway between source status and identification as a source, which is inappropriate. As methodologies of this sort are likely to become more common in future, this issue is a candidate for further investigation.

Clear from the between-subject relationship diagram (*Figure 6*) is the pivotal role played by the two long-term ICU residents, T012 and T126, as sources of colonisation. The former appears to be the originator of colonisations in at least two other subjects (and perhaps as many as six), and the latter three (up to five). When treating separate body sites as separate hosts (*Figure 7*), the direction of transmission was usually away from nasal colonisations. This is consistent with the role of the anterior nares as the ecological niche of *S. aureus* (*Kluytmans et al., 1997*). However, in our study the denser sampling from nasal sites likely also introduced some bias.

In over half of the subjects who screened positive for MRSA on more than one body site, there was no evidence for any site/phylogeny correlation. Some of the individual patient datasets involved were small and contained very few non-nasal isolates, in which cases the lack of detected structure may be due to lack of diversity. However, the correlation was absent even in subject T012, who was in the ICU throughout almost the entire study period, providing 223 isolates from three sites (and at least seven from each). Even for patient T126, where the phylogeny (see *Figure 8*) shows distinct axilla and endotracheal suction tube clades, other tips from those same sites appear distributed amongst nasal sequences. While further investigation of between-site colonisation using a sampling protocol designed specifically for the task would be advisable, these results suggest that the

establishment of a colony due to the transmission of a single lineage between sites, or the colonisation of distinct sites with distinct lineages, is quite rare, at least in a setting of this sort. It has been observed that eradication of nasal *S. aureus* frequently results in the disappearance of the bacterium from other sites (*Parras et al., 1995*; *Reagan et al., 1991*), and also that extranasal colonies are significantly less genetically diverse than nasal colonies (*Harkins et al., 2018*). Both observations suggest that the organism is readily and regularly spread from the anterior nares to other parts of the body, but that the resulting extranasal colonisations are frequently transient. Our results are consistent with this scenario.

While the bottleneck in colonisation of one body site from another is more often than not wide, that involved in subject-to-subject transmission was very variable amongst the five pairs examined. In one case only a single lineage appeared to be transmitted, while in others it was very wide indeed. This is perhaps unsurprising in a bacterial infection spread by contact, as the circumstances under which colonisation occurs, and the quantity of bacteria displaced, may vary greatly. There did not appear, in this small sample, to be any suggestion of an association between bottleneck width and the proximity of patient beds in an ICU. It appears that the size of the bottleneck can be very variable even in a single setting. This suggests that the shared genomic variant methodologies outlined by *Worby et al. (2017)* may be of limited use because they reconstruct sources of infection only infrequently when the bottleneck is tight, and that phylogenetic approaches that analyse multiple isolates per individual, of the type implemented in *phyloscanner*, are to be preferred as they enable reconstruction regardless of the bottleneck size. Short-read deep sequencing, however, is unsuitable for the reconstruction of within-host phylogenies in bacteria as their slow mutation rates do not provide sufficient phylogenetic resolution in segments of the genome of such a short length, while the approach of sequencing multiple colony picks that we employ here may be unsuitable for routine deployment for cost reasons. Ideally, a full bacterial genome could be sequenced without the intermediate step of growing the organism in culture, an approach which could be possible using single-cell genomics.

The SNP cut-off approach to identifying pairs has already been identified as potentially inadvisable; *Köser et al. (2012)*, in investigating a British neonatal outbreak, identified one outbreak strain of seven as a 'hypermutator' whose genetic distance from the remainder was an outlier. Nevertheless, not every investigation will have access to the means to do anything else. If sensitivity and specificity are considered equally important, then we recommend 39 SNPs as a threshold, which was suggested by both our gold standard sets. We would caution, however, that positive results are liable to lack predictive power at any SNP threshold in virtually any population survey unless the probability of a false positive test is extremely low. This is because, of the set of all pairs of subjects, the great majority will not be transmission pairs. In the ideal situation of a completely sampled transmission chain of $n$ individuals, only $n$-1 of the $n(n$-1)/2 pairs are transmission pairs, and for a positive test to be even 50% predictive of a positive result requires a false positive rate of at most 2/($n$-2). In large studies picking a threshold to fulfil this may in turn require unacceptably low sensitivity, and it should be noted that this is the best case scenario, in which every transmission pair is sampled. Ideally, sampling of within-host diversity and interrogation of the phylogenetic topology would be preferred to the use of a crude threshold, if this is feasible in a particular setting and using available resources.

A clear limitation of these results is their applicability to other settings, particularly to hospitals with greater (or poorer) resources for infection control. Better procedures would be expected to decrease the size of within-hospital transmission clusters, and also decrease the size of the bottleneck between both patients and body sites. In addition, our sampling was highly biased towards nasal isolates. We also lacked access to isolates from the general, extra-hospital, population that might have been used for phylogenetic comparison purposes. Finally, the sequencing protocol used here is expensive and replicating it will not be feasible in many settings.

This study demonstrates the additional insights into bacterial transmission that may be gleaned when sequencing multiple isolates per host. This is obviously essential when considering questions of bottleneck size and phyloanatomy, but also offers advantages in investigating the direction of between-host transmission. As costs decrease, it may become increasingly feasible to design and conduct studies with sampling frames that accommodate multiple sampling, and also to use it for investigations in clinical practice.

## Materials and methods

### Sequencing

DNA was extracted, libraries prepared and 100 bp paired end sequences determined for 2320 isolates on an Illumina HiSeq2000, as previously described (*Reuter et al., 2016*).

### Tagged genomic library preparation and DNA sequencing

Illumina reads were mapped onto the TW20 (accession number FN433596) reference sequence using SMALT (http://www.sanger.ac.uk/resources/software/smalt/) as previously described (*Tong et al., 2015*). A minimum of 30x depth of coverage for more than 92% of the reference genomes was achieved for both references (see Supplemental Table 1 of Tong et al). The default mapping parameters recommended for reads were employed, but with the minimum score required for mapping increased to 30 to make the mapping more conservative. Candidate SNPs were identified using samtools mpileup (*Li et al., 2009*) with SNPs filtered to remove those at sites with a mapping depth less than five reads and a SNP score below 60. SNPs at sites with heterogeneous mappings were be filtered out if the SNP is present in less than 75% of reads at that site (*Harris et al., 2010*). Identification of the core genomes was performed as previously described (*Holden et al., 2013*; *Harris et al., 2010*). The coordinates of the accessory regions of the TW20 chromosome, which were removed from the alignment for all later analyses, are described in *Supplementary file 2*. Recombination was detected in the genomes using Gubbins (http://sanger-pathogens.github.io/gubbins/) using the default parameters (*Croucher et al., 2015*).The predicted recombination regions of the TW20 chromosome are described in *Supplementary file 3*. Regions identified as the location of recombination were also removed from the alignment for all later analyses. De novo assembly of genomes of all isolates was performed using Velvet v0.7.03 (*Zerbino, 2010*).

### Phylogenetic reconstruction

The posterior set of phylogenies for the full alignment was generated using ExaBayes 1.5 (*Aberer et al., 2014*). The TW20 reference sequence was included as an outgroup. Due to computational memory limits, the alignment was divided into six sequential partitions of around 500,000 bp each, but in ExaBayes all parameters and the phylogenetic topology were linked across the six. Four MCMC runs, each in turn consisting of four coupled chains, were run for 5,000,000 generations with chain swaps every five generations and the heat factor set to 0.016. 50% of states in each run were discarded as burn-in, and the topologies forming output for all four runs were combined into a single tree set. The concatenated parameter trace for all four was examined to verify that the effective sample size (ESS) for the prior, likelihood and all numerical parameters was at least 200. The estimated ESS for the topology for the unified set of trees was calculated using RWTY 1.0.1 (*Warren et al., 2017*) and similarly verified to ensure a value of at least 200. The consensus tree was a 50% majority rule tree, with branch lengths, constructed using the *sumt* command in MrBayes 3.2.6 (*Ronquist et al., 2012*).

Input trees for the BaTS body site analysis were constructed by making separate alignments of all the sequences from each subject along with the TW20 outgroup. ExaBayes was run again on each with configurations varying depending on the number of sequences involved. Once again the output was checked to ensure all ESSs, including the estimated ESS for the topology, were at least 200, and consensus trees were built using *sumt*.

### Phyloscanner

A random sample of 100 trees from the ExaBayes posterior were used as input for *phyloscanner* v.1.4.2 (*Wymant et al., 2017*). Identification of subjects who experienced clear multiple colonisation was performed in a pre-processing step using *phyloscanner*'s dual infection detection utility with a *k* value of 30,432.1. For a single tree, this will divide the tips from a single subject into separate colonisations if the patristic distance between the MRCA nodes of each colonisation is equal to or greater than 100 SNPs. For each subject, the most common division when this procedure was applied to all 100 trees was used to define the number of colonisations and the sequences making up each one; this is equivalent to dividing a host's tips into two groups in cases where the median patristic distance between these MRCAs across the posterior was at least 100 SNPs.

*Phyloscanner* was then run in full on both the consensus phylogeny and the posterior set of trees. Four tips were identified by manual inspection as probable contaminants, due to suspicious genetic similarity to sequences from isolates from other subjects which were processed at the same time, and blacklisted. The tips coming from subject T126 were randomly reduced so that only ten per site per sample date were used, with the remainder also being blacklisted. The ancestral state reconstruction used *phyloscanner*'s modified Sankoff algorithm, with the $k$ parameter again set to 30,432.1. In the main analysis all tips from the same colonisation was given the same 'host' state in the reconstruction, but it was also repeated, further separating the set of tips by body site of origin. The collapsed tree from the results of *phyloscanner* applied to the consensus phylogeny was used to identify transmission events and plot these on the timeline of patient ICU stays and sampling events (*Figure 4*).

In the secondary analysis to investigate the effects of reducing the variation in tip counts between subjects on the results, *phyloscanner*'s downsampling tool was used to randomly reduce the tip counts for each subject to a maximum of five, and the analysis run again.

The cluster diagrams were created by defining two colonisations to be related in a single tree if they had at least one pair of adjacent subgraphs and all subgraphs from the pair were arranged in the 'ancestry', 'multiple ancestry' or 'complex' configurations (see *Wymant et al., 2017*). Departing slightly from default *phyloscanner* settings, which are designed primarily for HIV, we did not define relationships if the configuration was 'no ancestry' (this occurs where the reads from the two colonisations form sibling clades) and also did not use a patristic distance threshold. Edges were drawn between colonisations if they were related in at least 50% of posterior trees, and these edges used to identify the clusters.

When defining gold standard transmission pairs and non-transmission pairs for the SNP threshold analysis, both versions used posterior support of less than 0.05 for adjacency and a topological relationship of 'ancestry', 'multiple ancestry' or 'complex' to define the latter. The first version used posterior support of 0.95 of the same relationships to define pairs, whereas the second used support of 0.95 for adjacency and either 'multiple ancestry' or 'complex'.

## Phylogeny/body site association

The separate BaTS analysis (*Parker et al., 2008*) was performed on the posterior tree set constructed from the sequences from each patient whose isolates were obtained from more than one body site. In each such set, the TW20 outgroup was pruned from every tree before BaTS was run. 1000 replicates of state randomisation were used to estimate the null distribution of the AI, PS and MC statistics.

## Visualisation

Phylogeny diagrams were created using *ggtree* 1.8.2 (*Yu et al., 2017*).

## Acknowledgements

We thank Thibaut Jombart and Anne Cori for assistance in designing the sampling frame for the study, and Alice Ledda and Xavier Didelot for preliminary analyses.

## Additional information

### Funding

| Funder | Grant reference number | Author |
| --- | --- | --- |
| Wellcome | 098051 | Matthew TG Holden<br>Sharon J Peacock |
| Chief Scientist Office | SIRN10 | Matthew TG Holden |
| Wellcome | 106698/Z/14/Z | Vanaporn Wuthiekanun |
| Medical Research Council | G1000803 | Sharon J Peacock |
| European Research Council | PBDR-339251 | Matthew D Hall<br>Christophe Fraser |

The funders had no role in study design, data collection and interpretation, or the decision to submit the work for publication.

## Author contributions

Matthew D Hall, Conceptualization, Software, Investigation, Visualization, Methodology, Writing—original draft, Writing—review and editing; Matthew TG Holden, Resources, Data curation, Writing—review and editing; Pramot Srisomang, Investigation, Project administration, Writing—review and editing; Weera Mahavanakul, Investigation, Project administration; Vanaporn Wuthiekanun, Resources, Writing—review and editing; Direk Limmathurotsakul, Resources, Data curation, Supervision, Investigation, Methodology, Project administration, Writing—review and editing; Kay Fountain, Writing—review and editing, Comments and consultation when this work was first presented at a meeting resulted in very significant modifications to the interpretation of the analysis; Julian Parkhill, Conceptualization, Resources, Supervision, Funding acquisition, Project administration; Emma K Nickerson, Data curation, Writing—review and editing; Sharon J Peacock, Conceptualization, Resources, Supervision, Funding acquisition, Project administration, Writing—review and editing; Christophe Fraser, Conceptualization, Resources, Formal analysis, Supervision, Funding acquisition, Methodology, Writing—original draft, Project administration, Writing—review and editing

## Author ORCIDs

Matthew D Hall  https://orcid.org/0000-0002-2671-3864
Matthew TG Holden  http://orcid.org/0000-0002-4958-2166
Direk Limmathurotsakul  http://orcid.org/0000-0001-7240-5320
Kay Fountain  http://orcid.org/0000-0002-9984-5702

## Ethics

Human subjects: Ethical approval was obtained from the Ethical and Scientific Review subcommittee of the Royal Thai Government Ministry of Public Health (85/2550), and the Oxford Tropical Research Ethics Committee (024 07). All patients admitted to the two ICUs were eligible for inclusion and were enrolled after written informed consent, and consent to publish, was obtained.

## Decision letter and Author response

Decision letter https://doi.org/10.7554/eLife.46402.sa1
Author response https://doi.org/10.7554/eLife.46402.sa2

## Additional files

### Supplementary files

• Supplementary file 1. Examples of arrangements of tips with two states (red and blue) on a phylogeny and the corresponding values of the AI, PS, and MC statistics. Given are the values of the statistic and $p$-values estimated by permuting the tip states 1000 times. (**A**) Blue tips occur only in one clade but that clade is not exclusively blue. The AI statistic shows significant deviation from the null hypothesis of no phylogeny-state association, but the others do not. (**B**) Blue tips form a single clade. All statistics other than the MC for the red state show significant deviation from the null. (**C**) Blue tips appear randomly in the phylogeny. There is no evidence of any association using any statistic. (**D**) A single blue tip is basal to the remainder of the tree. The AI statistic and the MC for the *red* state suggest an association. (**E/F**) A single blue tip occurs amongst the red tips. Some evidence of association may still be identified using the AI statistic and further investigation is warranted. In cases D-F the PS and MC (blue) statistics are always equal to 1.

• Supplementary file 2. Coordinates of the accessory regions of the TW20 chromosome.

• Supplementary file 3. Coordinates of regions of the MRSA genome in which recombination was identified by Gubbins. Positions are with respect to the TW20 reference strain.

• Transparent reporting form

## Data availability

Illumina read data and genome assemblies are available in the European Nucleotide Archive as part of the studies with accession numbers PRJEB2076, PRJEB2489, and PRJEB4140. We are unable to provide patient data, beyond that contained in the figure source data files, for reasons of confidentiality.

The following dataset was generated:

| Author(s) | Year | Dataset title | Dataset URL | Database and Identifier |
|---|---|---|---|---|
| Wellcome Sanger Institute | 2019 | Staphylococcus aureus ST239 diversity in Thailand | https://www.ebi.ac.uk/ena/data/view/PRJEB4140 | European Nucleotide Archive, PRJEB4140 |

The following previously published datasets were used:

| Author(s) | Year | Dataset title | Dataset URL | Database and Identifier |
|---|---|---|---|---|
| Wellcome Sanger Institute | 2012 | Evolution of MRSA During Hospital Transmission and Intercontinental Spread | https://www.ebi.ac.uk/ena/data/view/PRJEB2489 | European Nucleotide Archive, PRJEB2489 |
| Wellcome Sanger Institute | 2011 | Staphylococcus aureus ST239 diversity | https://www.ebi.ac.uk/ena/data/view/PRJEB2076 | European Nucleotide Archive, PRJEB2076 |

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
