## [Decision Letter]

Thank you for submitting your article "Improved characterisation of MRSA transmission using within-host bacterial sequence diversity" for consideration by *eLife*. Your article has been reviewed by two peer reviewers, and the evaluation has been overseen by a Reviewing Editor and Neil Ferguson as the Senior Editor. The following individuals involved in review of your submission have agreed to reveal their identity: Don Klinkenberg (Reviewer #1); Taj Azarian (Reviewer #2).

The reviewers have discussed the reviews with one another and the Reviewing Editor has drafted this decision to help you prepare a revised submission.

The reviewers agreed that this study is a step forward compared to earlier such analyses in that it makes use of multiple sequences from multiple samples per patient, from various body sites, and that it addresses questions that can only be addressed because of this intensive sampling scheme: direction of transmission based on the phylogeny only, width of the bottleneck at transmission, and transmission between body sites within patients. In addition, based on the identified transmission pairs and clusters, the paper investigates the use of a SNP threshold to detect pairs and clusters in settings with fewer sequences available.

As a whole, it is a valuable contribution to the field of outbreak analyses with pathogen sequence data. The major strengths of the analysis are:

1) The application of a novel analytical approach to an unmatched dataset (995 isolates from 55 patients), which allowed them to move beyond "proof of concept" pilot analyses or simulations.

2) In doing so, the paper is able to address (with supporting data/results) three major unknowns about *S. aureus* genomic epidemiology: (i) the range of transmission bottleneck sizes (at least as it relates to the healthcare setting), (ii) the sensitivity and specificity of using a SNP cut-offs to determine transmission events and (iii) the compartmentalization/ migration of *S. aureus* to various body sites during carriage.

3) The utility of the approach (and accompanying software) to investigating transmission. This level of dense within-host sampling has never been attempted. Considering this and the results, this study is a significant advance from the original analysis by Tong et al., 2015 as well as the PHYLOSCANNER manuscript by Wymant et al., 2017.

4) The characterisation of aspects of MRSA hospital dynamics that were previously unknown or only suspected based on other types of evidence, and as a showcase of methods that can be used with such data.

5) The manuscript is clearly written, and we very much appreciate the efforts made to conceptually explain some of the methods, notably Figure 3 and Supplementary file 1.

Essential revisions:

1) "median patristic distance" (first, please repeat the use of median in the Materials and methods section). How sensitive is the result to this choice? For instance, can't a simple hierarchical clustering algorithm be used? Also, if additional samples are taken from a patient, the distance of these samples to the sequence(s) at infection will increase; consequently, the median distance to another colonisation should increase with larger clusters. Wouldn't it therefore be better to use the minimum distance?

2) Subsection “Trace colonisations”: Were the 9 single-swab colonisations with only one examination similar to the 24, e.g. in Figure 2? These could have been true colonisations, after all.

3) Figure 2 displays the median number of SNPs, but as explained above in comment 1, just increasing the number of samples is expected to increase this median. That would definitely cause bias in the comparison in Figure 2 and the Mann-Whitney U test.

4) Figure 2 is visually a bit misleading because of the 6 samples that are not in the Figure, four of which should be blue. Maybe add them at the right, below a ">100" mark on the X-axis?

5) The arguments for the “six trace colonisations” is strong, but there are 27 more. Do the data for the other 27 not suggest the same, or is it impossible to say for the other 27?

6) In the end, it seems reasonable to focus on the non-trace colonisations for the cluster and bottleneck analyses. However, for the SNP threshold analysis it should be realised that more than half of the cases were left out, so that identified transmission pairs could very well have been separated by intermediate cases. That affects the "gold standard" transmission pairs set.

7) Subsection “Reconstructed transmission events”, last paragraph: The nature of multiple arrows between cases should be explained. From the caption it appears that each arrow is associated with a unique subclade, but it is possible to conclude that multiple arrows between hosts indicate multiple lineages being transmitted, and thus a wide bottleneck?

8) Figure 4 legend: One aspect of the analysis that was not clear was if or how the time of transmission was estimated. In at least one instance, it appears that multiple transmission events occurred between two patients. From the text it states that "the timings represent […] the earliest time of sampling of all the sequences involved in the recipient subgraphs." Is this an assumption made for simplicity or is this method able to determine more accurately the timing of the event? This would be good to clarify because it has implications for the generalizability of the method to settings where the exposure period (in this case from admission to discharge) is not as well defined.

9) Subsection “Epidemiologically plausible transmissions have high posterior support”: It's difficult to understand why downsampling results in more impossible events. Wouldn't fewer samples per patient results in fewer "ancestry" configurations and more "no ancestry" configurations, so more external sources rather than (incorrect) internal sources? Is it possible to explain this phenomenon in one or two sentences?

10) Subsection “Transmission clusters”: A transmission cluster would be a set of cases that is linked by direct transmission, and separated from other clusters by unobserved transmission links. You define transmission clusters with phyloscanner based on "ancestry", "multiple ancestry" and "complex" configurations of the phylogenetic tree, and exclude the "no ancestry" configuration. From this it appears that you would exclude the relation between A and D in Figure 3 but it's unclear why: Figure 3 shows the singleton D is linked to A and not to B and C, which is a transmission link, even more so because transmission links are already defined with the possibility of intermediary patients. The text states "colonisations not closely related", which can't be concluded about A and D in Figure 3. Isn't it possible to connect the clusters of Figure 6 based on being siblings in the phylogenetic tree? Or is the variation between the 100 sampled trees too large to make this useful?

The problem is not so much in this part of the paper, which just describes the results following from the definitions. The problem is more in the SNP threshold part, where these defined clusters are used as gold standards for defining clusters with SNP thresholds. For that, it should be made clear why these are indeed separate clusters, and why they are set apart from one another.

11) Absence/presence of arrows in Figures 6 and 7: when comparing Figures 6 and 7, absence and presence of arrows is often not the same. For instance, in Figure 6 there are no arrows between C183-C249 and between C194-C249, but in Figure 7 they both point towards C249. In general, Figure 7 seems to have many more directions resolved than Figure 6. Suggestions: first, explain one example to provide intuition for the difference; second, make directions bidirectional (or draw two arrows) if there is evidence for transmission either way (back and forth).

12) Subsection "Between-subject transmission bottlenecks are of very variable size":

The transmission bottleneck size results are extremely valuable as these results are of great interest to multiple fields. Though, it is hard to judge how generalizable the bottleneck size findings are to other settings. While the authors are cautious in stating this, it would be beneficial to put their findings in the context of previous estimates. Further, they currently present the bottleneck size as the estimated number of transferred lineages, which will depend on the number of sequenced pathogens. Can this also be presented in terms of nucleotide diversity of the transmitted population?

13) "narrow bottleneck", but still two lineages at least, so why is that not wide?

14) Subsection “Between-subject transmission bottlenecks are of very variable size”: in discussing T183 and T194, the link with T178 is included, but considering Figures 6 and 7, C249 appears even more interesting, as that forms a triangle with both C183 and C194 (but with arrows in Figure 7)

15) Subsection “The majority of subjects showed no evidence of phylogeny/body site correlation”, last paragraph: subject T232 is mentioned with evidence for separate colonisation events, but in Figure 7 there is an arrow with equal support for the direction as for the link itself. That is not intuitive.

16) Subsection “The majority of subjects showed no evidence of phylogeny/body site correlation”, last paragraph: Figure 7: C178 would be expected to have multiple colonisation events, because the nose and throat samples are at completely different parts of the graph. That one missing here (and from Table 2) is also not intuitive.

17) Subsection “Identification of transmission pairs from single genomes”: The problem addressed is very relevant, but the gold standard definitions are not convincing.

First, transmission pairs (true positives) are defined by a posterior support of 0.95, but what about the set of no-transmission pairs? Is that the complete matrix of all other possible pairs in the dataset? What about the lower-support pairs, and the siblings (A and D in Figure 3)? If included into the set of no-transmission pairs, this set will contain transmission pairs as well.

Second, transmission pairs from the phyloscanner analysis can be through an intermediary hosts, especially because more than half of the positives (the trace colonisations) are left out. So, a transmission pair may not be a true transmission pair

Third, the definition of a cluster, only consisting of proven ancestral relations. As argued above, siblings in the phylogenetic tree may also be related through direct transmission, only without evidence of the direction. In the current analysis, these are treated as separate clusters.

18) Regarding the single genome sensitivity and specificity – The authors select one pair of genomes at random from all the possible pairs. It seems more practical to select the first isolate identified during sampling as this would be consistent with practice – most facilities conducting either active or passive surveillance usually don't re-swab after a positive surveillance swab is identified.

19) Discussion, eighth paragraph: As you state, it is currently not feasible to sequence 223 isolates from a single individual. In their analysis, they pool the samples from multiple time points and also down-sample to five isolates. It would be of great interest for the field to know what number of isolates is required to achieve this level of transmission/cluster prediction using this approach. What we imagine is presenting the sensitivity/specificity with progressive down-sampling from the entire sample and/or from different epochs. Are five random isolates from a single collection time point sufficient to replicate the accuracy of their analysis? Are multiple isolates longitudinally collected more important?

---

## [Author Response]

Essential revisions:1) "median patristic distance" (first, please repeat the use of median in the Materials and methods section).

We have added text to the Materials and methods to clarify this (subsection “Phyloscanner”).

How sensitive is the result to this choice?

This threshold was selected as a number that split up what seemed obvious instances of multiple colonisation by eye, and is not at all sensitive to variations within a reasonable range.

For instance, can't a simple hierarchical clustering algorithm be used?

Absolutely – this tool was used just because it is already included in *phyloscanner* and would give similar results to many other simple approaches.

Also, if additional samples are taken from a patient, the distance of these samples to the sequence(s) at infection will increase; consequently, the median distance to another colonisation should increase with larger clusters. Wouldn't it therefore be better to use the minimum distance?

We apologise; the text was somewhat misleading. The median distance is, as per *phyloscanner* convention, between the MRCAs of each clade, not between tips. It thus should be at most the minimum distance between tips. (The median is across the ExaBayes posterior, not the tip set.) This has been clarified (subsection “Identification of potential multiple colonisation events”).

2) Subsection “Trace colonisations”: Were the 9 single-swab colonisations with only one examination similar to the 24, e.g. in Figure 2? These could have been true colonisations, after all.

The number is actually 8 rather than 9, and we apologise for the typo. There is scant evidence of a difference in distributions between those 8 and the rest according to the metric in Figure 2 (*p*=0.164) or indeed between the 13 trace colonisations with a subsequent negative swab and those without (*p*=0.327). We acknowledge that some trace colonisations may indeed be genuine and the choice to exclude them all is a cautious one (subsection “Trace colonisations”).

3) Figure 2 displays the median number of SNPs, but as explained above in comment 1, just increasing the number of samples is expected to increase this median. That would definitely cause bias in the comparison in Figure 2 and the Mann-Whitney U test.

The reason for the use of the median here is that the null hypothesis is that an isolate from a trace colonisation is no different from a typical isolate from a non-trace colonisation. It would represent a single draw from the same distribution that produces the many samples from the latter. Hence we take the isolates from non-trace colonisations with the median genetic distance as representing “typical” examples. We have clarified the text (subsection “Trace colonisations”).

We do, however, acknowledge that there is a potential *temporal* source of bias with this approach. Isolates from trace colonisations cannot be acquired at anything but the first possible opportunity, whereas some from non-trace colonisations may have undergone further days or even weeks of mutation. If instead we take the median distance to the most similar earlier sample from amongst the isolates acquired on the first sample date, we see a similar tendency for shorter distances amongst trace colonisations, but it is not significant at the *p*=0.05 level (*p*=0.119). This can be seen in the new Figure 2—figure supplement 1.

4) Figure 2 is visually a bit misleading because of the 6 samples that are not in the Figure, four of which should be blue. Maybe add them at the right, below a ">100" mark on the X-axis?

This figure has been revised to switch to a log scale at distances beyond 50bp.

5) The arguments for the “six trace colonisations” is strong, but there are 27 more. Do the data for the other 27 not suggest the same, or is it impossible to say for the other 27?

This is not suggested by the data in the other cases. However, while such a pattern implies transient colonisation, not all transient colonisations would be expected to show it; such an event need not happen in the early part of a patient’s ICU stay. The only information we really have that strongly argues against transient colonisation (or indeed contamination) is repeated isolation of the same strain. We choose to emphasise examples where we have this, out of an abundance of caution. We have revised the text somewhat to clarify this point (subsection “Trace colonisations”).

6) In the end, it seems reasonable to focus on the non-trace colonisations for the cluster and bottleneck analyses. However, for the SNP threshold analysis it should be realised that more than half of the cases were left out, so that identified transmission pairs could very well have been separated by intermediate cases. That affects the "gold standard" transmission pairs set.

The trace colonisations are now present in one of the two new versions of the SNP analysis; they had to be excluded from the other for reasons that are outlined below.

7) Subsection “Reconstructed transmission events”, last paragraph: The nature of multiple arrows between cases should be explained. From the caption it appears that each arrow is associated with a unique subclade, but it is possible to conclude that multiple arrows between hosts indicate multiple lineages being transmitted, and thus a wide bottleneck?

This configuration does suggest transmission through a wide bottleneck, which we define as also encompassing a scenario involving multiple separate transmission events between the pairs (subsection “Between-subject transmission bottlenecks are of very variable size”). Text has been added to clarify this (subsection “Reconstructed transmission events.”).

8) Figure 4 legend: One aspect of the analysis that was not clear was if or how the time of transmission was estimated. In at least one instance, it appears that multiple transmission events occurred between two patients. From the text it states that "the timings represent […] the earliest time of sampling of all the sequences involved in the recipient subgraphs." Is this an assumption made for simplicity or is this method able to determine more accurately the timing of the event? This would be good to clarify because it has implications for the generalizability of the method to settings where the exposure period (in this case from admission to discharge) is not as well defined.

It was indeed merely for simplicity as we do not have a means to estimate such timings from the genetic information. This has been clarified (Figure 4 legend).

9) Subsection “Epidemiologically plausible transmissions have high posterior support”: It's difficult to understand why downsampling results in more impossible events. Wouldn't fewer samples per patient results in fewer "ancestry" configurations and more "no ancestry" configurations, so more external sources rather than (incorrect) internal sources? Is it possible to explain this phenomenon in one or two sentences?

The reason for this is that there is a tendency to infer the more frequently-sampled individual as the source in a likely transmission pair. Equalising counts reduces the topological signal in one direction and makes reconstructions of the opposite direction more frequent in the posterior set. This might seem at first glance to be simply a bias but, as we see here, matters are actually more complex. In our case removing this bias by downsampling results in a reconstruction which is plainly less accurate. High sample counts are not irrelevant information as they indicate something (a long ICU stays here) that in of itself makes an individual a more probable source. A high sample count lies at least partially on the causal pathway from a factor making source status more likely to detection as a source, and a variable lying on the causal pathway is not considered confounding. There is, nonetheless, undoubtedly scope for further investigation of this issue. An explanation of this has been added to the Discussion (fourth paragraph).

(This is a phenomenon which we have also seen in HIV, where viral load is associated with both high sample counts and more readily transmitting the virus.)

10) Subsection “Transmission clusters”: A transmission cluster would be a set of cases that is linked by direct transmission, and separated from other clusters by unobserved transmission links. You define transmission clusters with phyloscanner based on "ancestry", "multiple ancestry" and "complex" configurations of the phylogenetic tree, and exclude the "no ancestry" configuration. From this it appears that you would exclude the relation between A and D in Figure 3 but it's unclear why: Figure 3 shows the singleton D is linked to A and not to B and C, which is a transmission link, even more so because transmission links are already defined with the possibility of intermediary patients. The text states "colonisations not closely related", which can't be concluded about A and D in Figure 3. Isn't it possible to connect the clusters of Figure 6 based on being siblings in the phylogenetic tree? Or is the variation between the 100 sampled trees too large to make this useful?The problem is not so much in this part of the paper, which just describes the results following from the definitions. The problem is more in the SNP threshold part, where these defined clusters are used as gold standards for defining clusters with SNP thresholds. For that, it should be made clear why these are indeed separate clusters, and why they are set apart from one another.

See below for a full description of the changes to this analysis.

11) Absence/presence of arrows in Figures 6 and 7: when comparing Figures 6 and 7, absence and presence of arrows is often not the same. For instance, in Figure 6 there are no arrows between C183-C249 and between C194-C249, but in Figure 7 they both point towards C249. In general, Figure 7 seems to have many more directions resolved than Figure 6. Suggestions: first, explain one example to provide intuition for the difference; second, make directions bidirectional (or draw two arrows) if there is evidence for transmission either way (back and forth).

The updated versions of the network figures now have bidirectional arrows, and text has been added to explain the case of C249, which is the most striking example of the improved resolution in Figure 7 (subsection “Transmission clusters”).

12) Subsection "Between-subject transmission bottlenecks are of very variable size":The transmission bottleneck size results are extremely valuable as these results are of great interest to multiple fields. Though, it is hard to judge how generalizable the bottleneck size findings are to other settings. While the authors are cautious in stating this, it would be beneficial to put their findings in the context of previous estimates.

To our knowledge, there are no previous estimates of transmission bottleneck size between human subjects in the literature.

Further, they currently present the bottleneck size as the estimated number of transferred lineages, which will depend on the number of sequenced pathogens. Can this also be presented in terms of nucleotide diversity of the transmitted population?

We have added these numbers to Table 1 and discuss them (subsection “Between-subject transmission bottlenecks are of very variable size”).

13) "narrow bottleneck", but still two lineages at least, so why is that not wide?

We have revised the text here (subsection “Between-subject transmission bottlenecks are of very variable size”) in order to talk about “narrow” and “wide” in more relative terms.

14) Subsection “Between-subject transmission bottlenecks are of very variable size”: in discussing T183 and T194, the link with T178 is included, but considering Figures 6 and 7, C249 appears even more interesting, as that forms a triangle with both C183 and C194 (but with arrows in Figure 7).

This was overlooked because of the lack of significant overlap with the other patients in the same bed. However, T183 did briefly occupy a bed that had previously been occupied by T249, which we now mention (subsection “Between-subject transmission bottlenecks are of very variable size”).

15) Subsection “The majority of subjects showed no evidence of phylogeny/body site correlation”, last paragraph: subject T232 is mentioned with evidence for separate colonisation events, but in Figure 7 there is an arrow with equal support for the direction as for the link itself. That is not intuitive.

This part of the manuscript was too cursory and relied entirely on examining the consensus trees. We have changed it to a description of the patterns over the full posterior (subsection “The majority of subjects showed no evidence of phylogeny/body site correlation”). T232 has a quite poorly resolved topology which can support multiple scenarios, and in the main analysis the direction depicted in the original version of Figure 7 was the most well supported by the rules used to construct the diagram.

16) Subsection “The majority of subjects showed no evidence of phylogeny/body site correlation”, last paragraph: Figure 7: C178 would be expected to have multiple colonisation events, because the nose and throat samples are at completely different parts of the graph. That one missing here (and from Table 2) is also not intuitive.

We neglected to mention that T178, while sampled from two body sites, provided only a single sequence from each, which makes investigation using BATS impossible. The same is, in fact, true of T322. We have added text to this effect (subsection “The majority of subjects showed no evidence of phylogeny/body site correlation”).

17) Subsection “Identification of transmission pairs from single genomes”: The problem addressed is very relevant, but the gold standard definitions are not convincing.First, transmission pairs (true positives) are defined by a posterior support of 0.95, but what about the set of no-transmission pairs? Is that the complete matrix of all other possible pairs in the dataset? What about the lower-support pairs, and the siblings (A and D in Figure 3)? If included into the set of no-transmission pairs, this set will contain transmission pairs as well.Second, transmission pairs from the phyloscanner analysis can be through an intermediary hosts, especially because more than half of the positives (the trace colonisations) are left out. So, a transmission pair may not be a true transmission pairThird, the definition of a cluster, only consisting of proven ancestral relations. As argued above, siblings in the phylogenetic tree may also be related through direct transmission, only without evidence of the direction. In the current analysis, these are treated as separate clusters.18) Regarding the single genome sensitivity and specificity – The authors select one pair of genomes at random from all the possible pairs. It seems more practical to select the first isolate identified during sampling as this would be consistent with practice – most facilities conducting either active or passive surveillance usually don't re-swab after a positive surveillance swab is identified.

For the SNP analysis:

1) We have concluded that the definition of a transmission cluster that was used probably cannot be made entirely coherent, and have entirely removed that part of the paper.

2) Sequences for SNP analysis are now taken from just the first positive swab for each subject, rather than the pool of all sequences.

3) We now use a cut-off of 0.05 posterior support for any ancestry to define non-transmission pairs in the gold standard. Any pairs lying between 0.05 and 0.95 are not used.

4) As a result of modification 3, the PPV and NPV statistics become meaningless as these are setting-dependent and the study “population” is no longer even the complete set of pairs in our sample. We have removed these from the results. Nevertheless, the point that in most circumstances most pairs of subjects cannot be transmission pairs, and thus the PPV will be low for even medium-sized study populations unless the false positive rate is also low, is still valid, and we now make it in the Discussion by means of a mathematical sketch (Discussion, eighth paragraph).

5) When using *phyloscanner* as a gold standard for SNP distance thresholds, we could not use patristic distance thresholds as the reasoning becomes circular. In particular, the relationship between A and D in Figure 3 cannot be established as that of a transmission pair without reference to branch lengths, and branch lengths are themselves obviously very closely related to SNP distance. We used only the topological signal of transmission instead, and sibling clades are not a definitive signal of transmission.

6) The point about missing intermediaries is valid. To address this, we created a second gold standard set for which membership required posterior support of 0.95 for the “multiple ancestry” or “complex” topological relationships, which are equivalent to the “PP” configuration in Romero-Severson et al. That paper showed that PP tends to imply direct transmission. While the simulations in that paper were designed to be similar to HIV, the reason, that it is much more unlikely that two distinct lineages will be transmitted twice in succession, remains valid here (subsection “Identification of transmission pairs from single genomes”).

7) We agree that there is no particular reason to exclude trace colonisations from this analysis, but anything involving a trace colonisation cannot be used in the new gold standard of point 6 because it is impossible to identify multiple lineage transmission to a subject with a single tip in the phylogeny. As a result, we present both – the original version with trace colonisations included, and the new one with them excluded.

19) Discussion, eighth paragraph: As you state, it is currently not feasible to sequence 223 isolates from a single individual. In their analysis, they pool the samples from multiple time points and also down-sample to five isolates. It would be of great interest for the field to know what number of isolates is required to achieve this level of transmission/cluster prediction using this approach. What we imagine is presenting the sensitivity/specificity with progressive down-sampling from the entire sample and/or from different epochs. Are five random isolates from a single collection time point sufficient to replicate the accuracy of their analysis? Are multiple isolates longitudinally collected more important?

While this would be a very interesting addition, we feel that the paper is quite long as it is and would prefer to leave it for future work.